# Reaching culturally acceptable and adequate diets at the lowest cost increment according to income level in Brazilian households

Eliseu Verly, Jr[1]*, Nicole Darmon[2], Rosely Sichieri[1], Flavia Mori Sarti[3]

**1** Department of Epidemiology, Institute of Social Medicine, Rio de Janeiro State University, Rio de Janeiro, Brazil, **2** MOISA, Univ Montpellier, CIRAD, CIHEAM-IAMM, INRAE, Montpellier SupAgro, Montpellier, France, **3** Center for Research in Complex Systems Modeling, School of Arts, Sciences and Humanities, University of São Paulo, São Paulo, Brazil

\* eliseu.junior@uerj.br

## Abstract

### Objective

To identify food choices allowing the fulfillment of nutritionally adequate diets resembling actual food patterns at the lowest cost achievable for the Brazilian population, stratified by income level.

### Methods

Food consumption and prices were obtained from the Household Budget Survey (n = 55,970 households) and National Dietary Survey (n = 32,749 individuals). The sample was stratified into capitals of the states and further by income levels according to the official minimum wage (totaling 108 geographic-economic strata, or GES). Linear programming models were performed for each GES in order to find the lowest cost of diets that meet a set of nutritional constraints. In order to find realistic diets, constraints referring to preferences were introduced in the models allowing optimized food quantities to depart progressively from the current intake for each food and food group. The impact of meeting each target nutrient was assessed by performing models removing each nutrient at the time.

### Results

The observed and optimized diet costs were US$2.16 and US$2.58 per capita/day. The highest cost increment and the greatest food shifts were observed in the lowest income level. The nutrient adequacy was reached by mainly increasing fruits and vegetables, beans, fish and seafood, dairy, nuts, and eggs; and reducing red and processed meat, chicken, margarine and butter, cookies, cakes, sugar-sweetened beverages, and sauces. As the departure from the current intakes increase, the optimized healthy diet cost reduced. In the lowest income, the lowest cost increment was about US$ 0.10; in the higher income levels, it tended to be cheaper than the observed cost. Calcium was the most expensive nutrient to meet adequacy.

**Data Availability Statement:** Data are publicly available for downloading on the Brazilan Institute of Geography and Statistics website (https://www.ibge.gov.br/en/statistics/social/population/25610-

pof-2017-2018-pof-en.html?=&t=o-que-e).
Variable names, description, and contents are
Portuguese.

**Funding:** Financial Disclosure: EVJ Grant number:
E26/203.263/2017 Funder name: FUNDAÇÃO
CARLOS CHAGAS FILHO DE AMPARO À
PESQUISA DO ESTADO DO RIO DE JANEIRO
(FAPERJ) Funder URL:www.faperj.br The funder
did not play any role in the study design, data
collection and anaysis, decision to publish, of
preparation of the manuscript.

**Competing interests:** The authors have declared
that no competing interests exist.

## Conclusion

Nutritionally adequate diets are possible but costlier than the observed.

## Introduction

Nutrient intake recommendations, such as those from Dietary Reference Intakes, are used worldwide for several purposes in the nutrition field, especially for dietary planning and assessment [1], food guide development [2], and fortification and nutrition-related policies [3]. Studies from developed [4,5] and developing countries [6] have shown a high prevalence of inadequate intakes of some nutrients.

Food prices are one of the most important determinants of food choices, and cost constraints are well-known barriers to the adoption of healthy food choices in populations with low-socioeconomic status (SES) [7]. Early observational studies of dietary intake in Australia [8], UK [9], and France [10], have found that healthier diets tend to be more expensive than less-healthy diets, a finding later confirmed in the US [11] and worldwide [7,12]. In fact, when the budget for food is very low, it is logical to select foods that are cheaper sources of calories [13], but those foods generally have high energy density and low micronutrient density [14], leading to unhealthy dietary intakes [13,15].

Thus, if improvements of dietary intakes induce extra-cost to the consumer, they are not likely to be adopted in the population, especially in low SES populations, where a high percentage of the total income is already assigned to food purchase. For example, approximately 21% of households surveyed in Brazil within the Household Budget Survey 2008–2009 had a monthly income of less than two official minimum wages; in these households, the percentage of the total income spent on food was 27.8%, while this value for the whole population was 16.1% [16].

On the other hand, food consumption changes toward a healthy diet might be hampered by food preferences, even in the context where cost is not an important constraint on food purchasing. For example, it is a consensus from the literature that richer people have a diet with a higher amount of nutrients than have poorer ones, however, it does not ensure they have nutritionally adequate diets [17].

Linear programming (LP) is an analytic method for the optimization of variables, subject to constraints expressed as target values that should be met. It helps in the assessment of the feasibility of complex problems involving multiple variables and constraints (such as cost and nutrient contents) and to find their optimal solution [18]. Results from studies with diet optimization may support decisions on food policies and nutritional guidelines after considering constraints of cost and acceptability, indicating what could be more effective and efficient in terms of diet changes, i.e., higher quality and acceptability and lower cost. Diet optimization studies based on dietary data in France demonstrated that it is possible to decrease the cost of a nutritious diet at the expense of social acceptability [19]. Using linear programming, they found a range of cost for nutritionally adequate diets, but this resulted in a greater change in the observed diets when the budget was restricted, that is, the reduction of the distance between current consumption and the optimized food plans led to higher monetary costs.

While it is known from some studies in developed countries that nutritional adequacy can be reached with realistic modifications within the habitual food patterns of individuals at no or little cost increment [20], there is no evidence that it may apply to developing countries, especially for low-income subpopulations. In this study, we aimed at identifying the food choices allowing to fulfill nutritionally adequate diets that most resemble the actual food pattern at the lowest cost achievable for the Brazilian population, stratified by income level.

## Methods

### Surveys

We used data from the National Dietary Survey (NDS) and the Household Budget Survey (HBS), both conducted in 2008 and 2009 by the Brazilian Institute for Geography and Statistics. NDS was simultaneously collected in a random subsample of ~25% of the HBS. A two-stage sampling process was adopted: in the first stage, census tracts were randomly selected; and, in the second stage, households were randomly selected within census tracts. Census tracts (n = 12,800) were grouped into 550 household strata with geographical and socioeconomic homogeneity, and the number of tracts in each stratum was proportional to the number of households in the stratum. The samples included 55,970 households (HBS) and 13,569 households (NDS). Household visits in each stratum were uniformly distributed throughout the 12-month period to encompass seasonal variations in both food intakes and prices. More information on the surveys can be found elsewhere [21].

### Unit of analysis

There is large heterogeneity in the food patterns and prices throughout the macro and microregions of the country. Thus, instead of performing LP models taking into account the mean observed food consumptions and prices for the whole population, we developed LP models separately for several geographically delimited sampling strata, defined as follows: the 550 household strata were collapsed into 26 Brazilian States and one Federal District, and further stratified into income levels according to the *per capita* income: $\leq$0.5 official minimum wage (MW), >0.5 and $\leq$1.5 MW, >1.5 and $\leq$3 MW, and > 3 MW (Minimum wage: BRL415.00 (Brazilian Reais), equivalent to US$179.65 in January 2009), totaling 108 aggregated strata (named geographic-economic strata, or GES). This rearrangement was adopted to improve the precision of the food consumption and price estimates by increasing the number of households in each unit of analysis (i.e., each GES). Due to the long period of data collection, family income was adjusted to the same reference date (January 31$^{th}$ 2009) using official inflation rates (National Consumers' Prices Index) to allow comparability between households visited several months apart.

### Model inputs

**Dietary intakes.** Dietary intake based on the NDS was collected from two non-consecutive food records (97% response rate for the second food record) filled by 32,746 individuals $\geq$10 years old (pregnant and breastfeeding women excluded; n = 1,254). It was reported 305 different food items, most of them were aggregated into a single food, for example, different types of banana into banana, or different preparation of red meat (boiled, roasted, grilled, etc.) into red meat. The aggregation resulted in a list of 102 foods.

**Food prices.** Food prices were extracted from the HBS database, where all the household members registered the amount purchased and expenditure with each food product for home consumption over a one-week period. The purchase records (about 850.000) were registered in a specially designed booklet. Data referring to prices were indirectly inferred: individuals reported expenditures and amounts, and prices were calculated using the division of expenditure per item in relation to its respective amount, and then converted into prices per 100g of edible portion. The information on expenditures is collected using both self-reported information and receipts presented by the individuals interviewed, which are checked by the interviewer in order to ensure its reliability. Food product subtypes were clustered (e.g., different types of orange into 'orange') which resulted in the same 102 foods as described above. The

final price for each food was estimated as the mean price of the food subtypes weighted by the frequency of reporting in the budget survey.

We matched each food price to its corresponding food reported in the dietary intake survey according to the GES, thus, the price variation over the GES was preserved. Considering the variation in food prices throughout the period of collection, all prices were deflated to the same reference date (January 31[th] 2009) using official inflation rates.

**Mean observed food intakes and mean observed cost.** Mean food intakes were calculated for each GES. Overall mean food intakes, i.e., the mean intakes over all the 108 GES, are referred to as 'mean observed diet'. Likewise, overall mean diet cost (i.e., the mean diet cost from all the 108 GES) is referred to as 'mean observed cost'. We excluded non-food nutrient and energy sources from the food list, i.e., coffee and tea (without sugar), and alcoholic beverages, resulting in a list varying from 44 to 98 foods, according to the GES. The set of food declared as consumed in each GES is designed as the GES-specific food repertoire.

We calculated the nutrient content of the observed and optimized diets using the Brazilian Food Composition database (TBCA-USP) [22]. Nutrient composition of food subtypes clustered into foods (e.g. different types of rice into 'rice') was estimated as the mean composition of the foods weighted by the frequency of reporting in the dietary survey. The food consumptions estimated in each GES were used as the starting point for the optimized diets in the linear programming models.

## Linear programming models

A linear programming model is defined by an objective function which is optimized (i.e., minimized or maximized), depending on decision variables restricted by various constraints [19]. In the present study, decision variables were the quantities (in grams) of foods from the GES-specific food repertoire. Foods not reported to be consumed by any individual in a given GES were allowed to be introduced as decision variables when they were reported by individuals from another GES within the same state. We develop linear programming models to obtain the lowest cost of nutritionally adequate diets for each GES, with the objective function described as follow:

$$min\ y = \sum_{i=1}^{i=n} (Q_{f,g}^{opt} \cdot price_{f,g}) \tag{1}$$

Where $y$ represents the objective function to be minimized, $i$ is the food reported in the GES $g$ from a set of $n$ foods consumed, $Q_{f,g}^{opt}$ and the $price_{f,g}$ are, respectively, the optimized quantity and the price of the food $f$ in $g$.

**Models constraints.** The realism of the optimized diet, defined as how much it resembles the current observed diet, was reached by imposing two sets of acceptability constraints as described below:

Acceptability food constraints: These are boundaries in which optimized quantities of foods may deviate from the observed mean intakes in order to avoid optimized diets being culturally or socially unacceptable. Acceptability constraints may include lower and upper values, that is, the lowest and highest amount for a given food allowed in the models. Some studies using linear programing have applied acceptability constraints derived from the population intake distribution, such as the 10[th], 20[th], 80[th] or 90[th] percentile of intake [19]. In Brazil, a previous study showed that meeting all nutrients is mathematically impossible considering boundaries for each food derived from the population intake distribution [23], meaning that, in order to fulfill all nutritional constraints, more flexible boundaries should be allowed. In order to find the

lowest deviation from the observed diets consistent with a feasible solution, and to assess the relationship between cost and acceptability, the boundaries were introduced in the models allowing optimized food quantities to vary progressively from the observed mean intake for each food. It was done by performing 300 models for each GES, imposing upper boundaries $ub$ that consist in $[m_{f,g} + d]$, where $m_{f,g}$ is the mean quantity of the food $f$ observed in a given GES $g$, and $d = (1,2,3, \ldots, 300)$ is the allowed deviation from the observed amount reported of each food $f$ (hereafter referred just as 'deviation $d$'). The constraint $ub$ for the food $f$ was, however, censored to its GES-specific current mean portion size, defined here as the mean consumption in a consumption-day. That is, the food $f$ content allowed in the optimized diet should not be higher than what people eat, on average, in the day when the food is eaten. The lower boundaries $lb$ were obtained from the equation ($[1/(ub/m_{f,g})] * m_{f,g}$). The rationale behind this equation is that the increase and decrease allowed for a given food is proportional to deviation $d$ (e.g., if a given food is allowed to double the amount, it will consequently be allowed to be reduced by half). Furthermore, it prevents the total removal of a given food from the GES-specific food repertoire because the $lb$ never gets to zero.

Acceptability food group constraints: Additional constraints were imposed on quantities from each food group. In each GES model, food group quantities were not allowed to be higher than a defined boundary $p$ to avoid unrealistic diets. These boundaries ($p$) corresponded to the mean food group portion size. The boundaries assigned to each GES were, however, established according to its corresponding state. In case of model unfeasibility for a given GES, we allowed both $d$ and $p$ to progressively increase by every one gram till the model finds a feasible solution.

Nutritional constraints: Constraints for calcium, magnesium, iron, phosphorus, copper, zinc, vitamins A, B1, B2, B6, B12, C, niacin, and folate were introduced in the models. Values were derived from the Estimated Average Requirement (EAR) [1]; as they are age-sex specific, the overall constraint for each nutrient corresponded to mean values of requirements (i.e., mean of age-sex EAR values) weighted by the frequency of age-sex group in the population. Nutrients without EAR were not constrained in the models. Total energy content was constrained to be equal to the mean estimated energy requirement (EER) [25] calculated using age, sex, and anthropometric information specific for each GES (mean EER over the 108 GES = 2,113 kcal). Due to the absence of information on the accuracy of estimates for added salt in food preparations, the ratio sodium/kcal in the optimized diet was constrained to be equal or lower than the ratio in the observed diet obtained for each GES. Saturated and *trans* fats and added sugar constraints were based on the WHO Report [25]. The whole set of nutritional constraints introduced in all of the models, including macronutrients, fats, and sugar, as well as the reference for each recommendation is presented in **Table 1**.

**Influence of the nutrient targets on cost and tolerability.** In order to assess the impact of meeting adequacy for each nutrient on the cost and the acceptability, the models were performed removing the constraint for one nutrient at a time. For example, say a model was estimated keeping all nutritional constraints but removing calcium constraint, then, performed again keeping all nutritional constraints (including calcium) but removing zinc constraint, and so forth for all nutrients. Linear programming models were performed using the Optmodel Procedure from software SAS OnDemand.

## Descriptive analyses

The impact of meeting nutritional adequacy on cost and food group changes was assessed by comparing the cost differences across the solutions for the range of deviation $d$ introduced progressively in the models. In this analysis, each GES could reach the nutritional adequacy at

**Table 1. Nutritional constraints imposed in the models.**

| Component | Constraint |
|---|---|
| Energy (kcal) | = EER[a] |
| Proteins | 10% - 35% kcal[a] |
| Carbohydrates | 45% - 65% kcal[a] |
| Total fats | 20% - 35% kcal[a] |
| Saturated fat | <10% kcal[b] |
| Trans fat | < 1% kcal[b] |
| Added sugar | <10% kcal[b] |
| Total fiber | $\geq$ 30g[c] |
| Sodium/kcal | $\leq$ observed ratio[d] |
| Calcium | $\geq$ 868mg[c] |
| Magnesium | $\geq$ 303mg[c] |
| Phosphorus | $\geq$ 649mg[c] |
| Iron | $\geq$ 6.8mg[c] |
| Copper | $\geq$ 0.7mg[c] |
| Zinc | $\geq$ 8mg[c] |
| Vitamin A | $\geq$ 560mg RAE[c,e] |
| Vitamin B1 | $\geq$ 0.9mg[c] |
| Vitamin B2 | $\geq$ 1mg[c] |
| Vitamin B6 | $\geq$ 1.1mg[c] |
| Vitamin B12 | $\geq$ 2mcg[c] |
| Vitamin C | $\geq$ 66.1mg[c] |
| Niacin | $\geq$ 11.5mg[c] |
| Folate | $\geq$ 322mcg DFE[c,f] |

[a] Estimated Energy Requirement [23].

[b] World Health Organization [24].

[c] Derived from the Estimated Average Requirement [1].

[d] Ratio sodium/kcal observed in each GES.

[e] Retinol Activity Equivalents.

[f] Dietary Folate Equivalents.

different deviations, resulting in different costs and food changes. Thus, for all the GES feasible solutions, we ranked the solutions in ascending order by the deviation *d* and *p*, so that we have the solution with the lowest deviation for each GES, and then the second-lowest deviation for each GES, and the third, and so on, till the *k* solutions. The mean cost and deviation over the 108 GES were computed for each *k* solution. The first *k* solution was that in which each GES reached the nutritional adequacy at the lowest deviation, then it was considered the best solution, that is, the one with the lowest cost at the lowest deviation from the observed diet.

The mean observed and optimized cost, mean cost difference, mean food changes, and mean deviation *d* were weighted by sampling weights and presented for the whole sample and stratified by income levels. The cost difference (*CD*) was calculated according to the following equation:

$$CD_g = \left( \sum_{i=1}^{i=n} price_{f,g} \cdot Q_{f,g}^{opt} \right) - \left( \sum_{i=1}^{i=n} price_{f,g} \cdot Q_{f,g}^{obs} \right) \tag{2}$$

Where: $price_{f,g}$, $Q_{f,g}^{opt}$, and $Q_{f,g}^{obs}$ are respectively the GES-specific price, the optimized quantity, and the observed quantity of the food *f* in the GES *g*.

We assessed the relationship between cost and deviation by plotting the cost difference with the mean deviation across the $k$ solutions, also stratified by income levels. We also plotted the changes in the food groups across the $k$ solutions (optimized diets that most resemble the observed diet) in order to assess the relationship between cost and deviation in the changes in food group quantities compared with the observed diet. Finally, we used the same procedure to select the best solution for the models assessing the impact of each nutritional constraint on the cost and deviation. In the result section, the 102 foods used in the LP models were grouped into a smaller number of foods or food groups according to the analysis. The food categories can be accessed on the S1 Appendix.

### Ethics

The protocol of this research was approved by the Ethics Committee of the Instituto de Medicina Social of the Universidade do Estado do Rio de Janeiro (CAAE 0011.0.259.000–11).

## Results

Most of the feasible solutions were achievable for the deviation $d$ (allowed deviation for each individual food amount in $g$) between 15g and 40g. In 98 out of 108 GES, feasible diets were achieved within the $p$ constraint introduced in the models (maximum allowed amount for each food group in each $g$). For the remaining GES, an average increase of 23g in $p$ was needed.

### Selected best solutions

For the selected best solutions, the sum of all the positive changes (increases in food quantities after optimization) was, in average over all the GES, 346g, and the sum of all negative changes (decreases in food quantities) was, in average over all the GES, 96g; the overall increase in the food quantities was, in average, 250g. The food repertoires after optimization varied from 52 to 102 foods over the 108 GES. Table 2 shows the mean cost of the observed and best-optimized diets, the mean cost differences, and the mean deviation $d$ over all the GES. Results are presented for Brazil and stratified by monthly per capita income in minimum wage.

The mean observed diet cost (per person/day) in the population was US$2.16 (BRL 4.99), ranging from US$1.84 (BRL4.25) (lowest income) to US$2.60 (BRL6.00) (highest income). The mean cost of the optimized diet was US$2.58 (BRL5.96), ranging from US$2.59 (BRL5.98)

**Table 2. Mean (standard error) observed diet cost, in US$, mean cost of optimized diets[a], mean cost difference and mean deviation $d$ (in g/d) from the observed diets, according to income level (n = 108 GES[b]).**

|  | Observed diet cost | Optimized diet cost | Cost difference[d] | Cost difference (%)[e] | $d$[f] |
|---|---|---|---|---|---|
| <0.5MW[c] | 1.84 (0.07) | 2.59 (0.11) | 0.74 (0.08) | 40 | 51.2 (19.9) |
| 0.5–1.5MW | 2.03 (0.03) | 2.57 (0.09) | 0.54 (0.08) | 26 | 36.9 (13.3) |
| 1.5-3MW | 2.29 (0.03) | 2.52 (0.05) | 0.22 (0.03) | 10 | 28.7 (8.3) |
| >3MW | 2.60 (0.02) | 2.64 (0.17) | 0.04 (0.19) | 1.5 | 41.9 (20.3) |
| Brazil | 2.16 (0.04) | 2.58 (0.05) | 0.41 (0.05) | 20 | 38.4 (7.91) |

[a] Lowest cost at the lowest deviation from the observed diet.

[b] Geographic-economic strata.

[c] Minimum wage.

[d] (optimized diet cost–observed diet cost).

[e] (optimized diet cost–observed diet cost) / optimized diet cost * 100.

[f] Mean of $d$ (acceptability food constraint introduced in the models).

in the lowest income to US$ 2.64 (BRL6.09) in the highest income. The highest cost increment for the nutritionally adequate was observed in the lowest income. The diet cost was almost similar to the observed cost in the highest income. Moreover, the deviation from the observed diets needed to reach adequacy at the lowest cost was higher in the lowest income level, compared with the other levels. The main changes to reach adequacy included an increase in beans, fish and seafood, dairy, nuts, and eggs; and a decrease in red and processed meat, chicken, margarine and butter, cookies, cakes, sugar-sweetened beverages, and sauces (Table 3).

## Cost versus deviation

Fig 1 represents the cost differences (i.e. change in cost after diet optimization) in relation to the deviation $d$ from the observed diet: each symbol represents the mean cost difference over the 108 GES according to the distance $d$; the solid curve is the predicted cost difference. The more realistic the diet (less deviation $d$), the higher the cost increment. But this relationship

**Table 3. Mean (standard error) food contents in the observed and optimized diets[a] (n = 108 GES[b]).**

| Foods / Food groups (grams/day) | Observed diet | Optimized diet |
|---|---|---|
| Legumes | 194.1 (9) | 215.9 (7) |
| Rice | 165.9 (6.1) | 166.6 (6.2) |
| Red meat | 87.7 (2.7) | 68.5 (3.1) |
| Chicken | 38.9 (1.3) | 26 (1.4) |
| Eggs | 12.3 (0.7) | 23.7 (1.4) |
| Fish / seafoods | 29.7 (3.9) | 42 (4.1) |
| Processed meats | 11.2 (1) | 5.3 (0.8) |
| Fruit | 216.3 (7.8) | 341.8 (12.8) |
| Leafy vegetables | 20.3 (1.3) | 27.9 (2.1) |
| Other vegetables | 41.3 (2) | 37.6 (2.4) |
| Nuts | 0.2 (0.02) | 10 (0.7) |
| Tuber | 40.2 (1.8) | 34.8 (2.3) |
| Whole cereals | 7.2 (0.8) | 9.2 (1.6) |
| Milk | 125.5 (3.9) | 148.9 (3.9) |
| Non-fat milk | 4.7 (0.6) | 39 (5.8) |
| Yogurt | 10.2 (1.2) | 47.5 (3.2) |
| Cheese | 8.4 (1) | 30.2 (1.5) |
| Oils | 7.1 (0.2) | 4.3 (0.4) |
| Olive Oil | 0.1 (0.01) | 0.1 (0.04) |
| Breads | 55.3 (1.6) | 66.3 (1.6) |
| Cake | 13.6 (0.5) | 6.7 (0.5) |
| Cookies | 15.9 (0.8) | 8.5 (0.7) |
| Pasta | 51.8 (2.1) | 67 (3.2) |
| Sauces | 20.3 (1.5) | 13.1 (1.4) |
| Manioc flour | 8.5 (1.9) | 21.2 (1.4) |
| Snacks | 22 (1.7) | 18.1 (1.7) |
| SSB[c] | 128.9 (8.5) | 109.4 (9.1) |
| Sweets | 37.1 (1.5) | 35.8 (0.9) |

[a] Lowest cost at the lowest deviation from the observed diet.

[b] Geographic-income strata.

[c] Sugar-sweetened beverages.

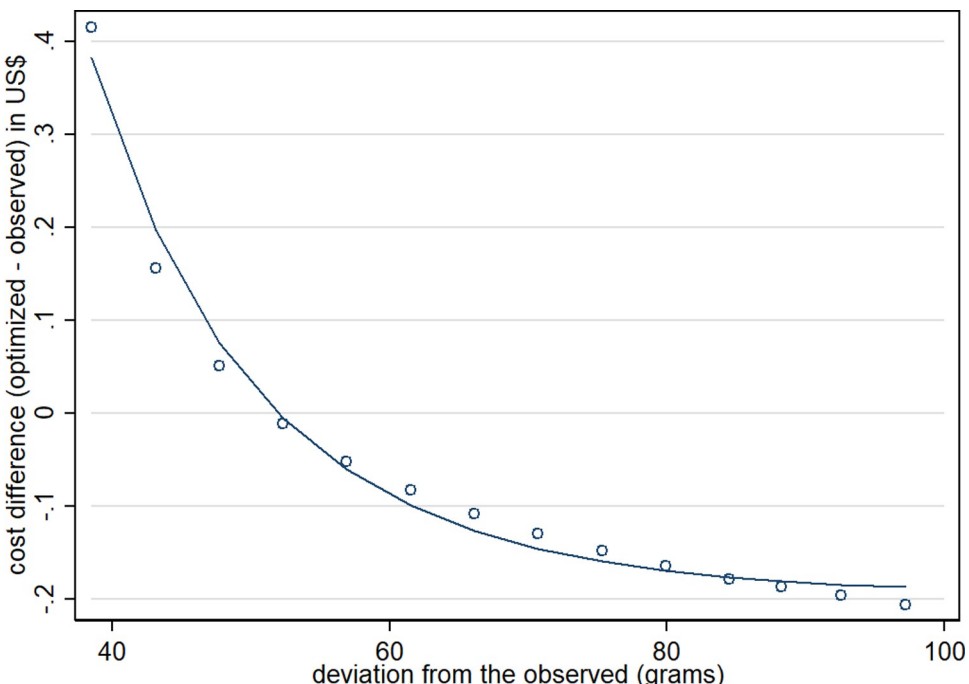

**Fig 1. Cost differences between optimized and observed diets, in relation to the deviation from the observed diet[a].**
[a] Mean of deviation *d* (acceptability food constraint introduced in the models). Each symbol represents the mean cost difference over the 108 GES according to the distance *d*; the solid curve is the predicted cost difference.

was not linear: the cost increment was very high in the lowest deviation *d*, then it decreased dramatically when increasing the deviation *d*, and tended to stabilize at a negative cost difference of about US$ 0.40 (small decreases in diet cost for very high deviation *d*). The main changes in the food quantities with deviance *d* are presented in **Fig 2**. The dashed lines represent the least change in each food group needed to move the observed diets toward nutritionally adequate diets. Further changes (keeping diet adequacy) are represented by the solid lines. The *x*-axis, from left to right, represents the increase in deviation *d*, and consequently the reduction in the diet cost. In general, the higher the deviation *d*, the lower the amount of fruit, vegetables, tubers and nuts, red and processed meats, chicken, fish and seafood, and ready-to-eat foods. These reductions are compensated by the increase in beans, rice and brown rice, dairy, margarine and butter, and refined grains.

The same pattern of a non-linear relationship between the cost difference and deviation from the observed consumption was observed when stratified by income level. The cost differences in the income levels higher than 0.5 MW per capita tended to values lower than zero with the increase in the deviance *d*. In the lowest income level, however, the cost increment tended to stabilize about US$ 0.10 (**Fig 3**).

## Impact of each nutrient adequacy on the cost and deviation

When performing the optimized models removing each nutritional constraint separately, all models presented the same or very similar results concerning both cost difference and deviation compared with the full model (model with all the nutritional constraints). The exception was for calcium, for which the cost difference was US$ -0.04 (BRL -0.11) and the mean deviation *d* was 14.7; lower values compared with US$ 0.41 (BRL 0.94) and 38.4 from the full model

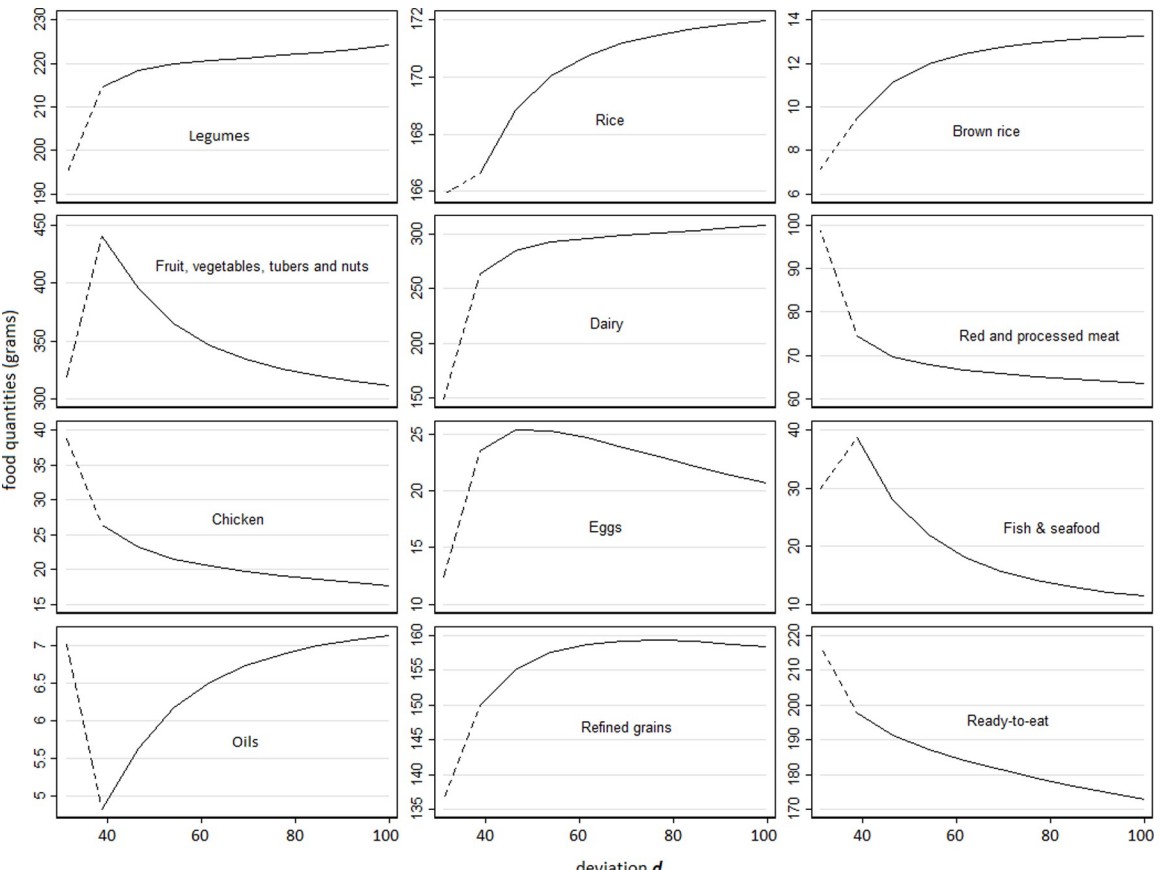

**Fig 2. Mean food group changes[a] across the solutions that most resemble the observed diet[a].** [a] Mean of deviation *d* (acceptability food constraint introduced in the models).

including the calcium constraint. However, the cost difference varied according to the income level: from US$ 0.18 in the lowest to US$ -0.39 in the highest income level. In general, the main impact on food quantities after relaxing calcium constraint, compared with the full model, were observed, cheese (-21g), yogurt (-43g), whole and non-fat milk (-61g), fruits (-86g), and fish and seafood (-21g). Mean calcium content in this set of models was 505mg.

## Discussion

In this study, we aimed at finding the lowest cost for the optimized nutritionally adequate diets that most resemble the actual food patterns in Brazil. Results are not intended to be seen as strict quantities that should be eaten by everyone, but as an indication of the main changes to improve diet quality at the lowest achievable costs.

Our results have some relevant implications. First, higher nutrient adequacy demands substantial changes in food choices. One of the main concerns when designing dietary recommendations and optimized diets is to ensure that it is locally acceptable. Unrealistic food quantities were prevented in this study by imposing acceptability boundaries ranging from more to less stringent constraints. However, it was possible to obtain solutions for most of the GES at acceptability boundaries near the mean food group portion size or less, and deviation from the observed intake between 15g and 40g. It means that each individual food was allowed to

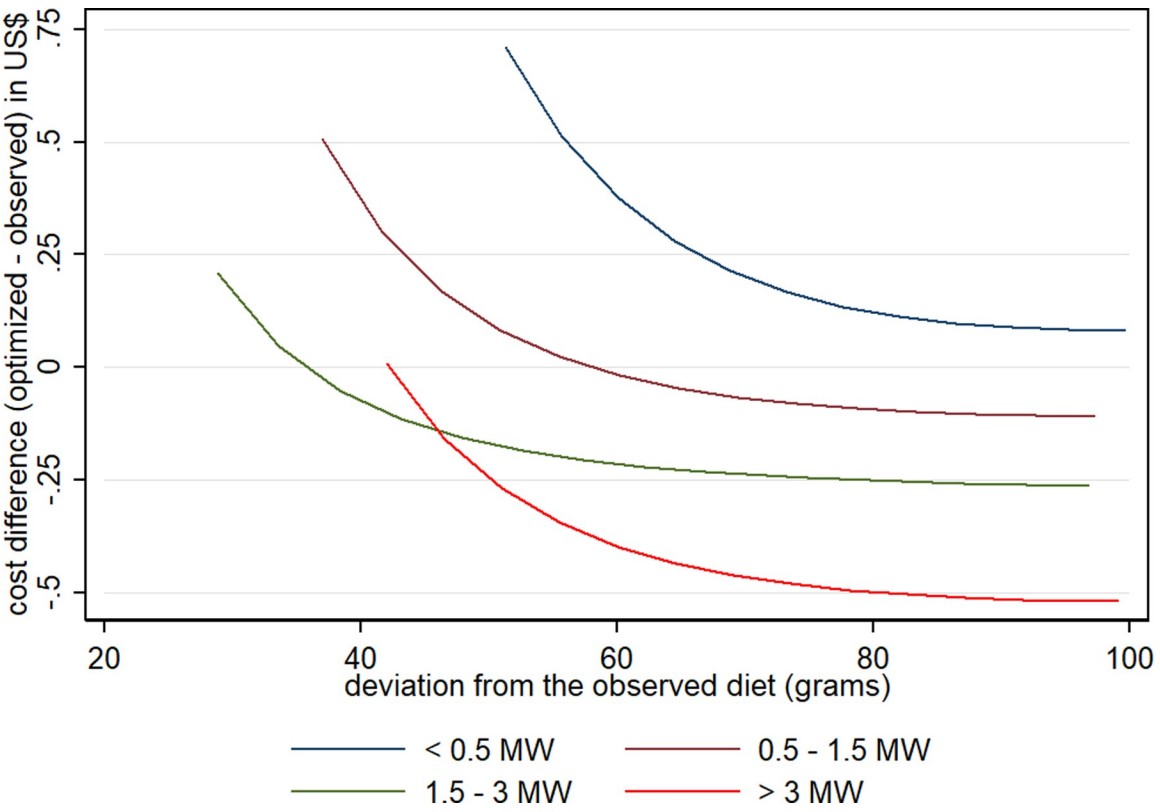

**Fig 3. Cost difference between optimized and observed diets, in relation to the deviation from the observed diet[a], for each income level (in MW[b]).** [a] Mean of deviation $d$ (acceptability food constraint introduced in the models). [b] MW = minimum wage.

increase up to 40g (the amount allowed to decrease depends on the observed intake quantity), which led to a net overall increase, on average over the 108 GES, of 250g/day.

Our results showed the most realistic diet within acceptable cost and preference to meet nutritional adequacy, although the optimization has resulted in a considerable distance from the actual consumption for some foods. For example, beans were increased by 21g/day to reach nutritional adequacy, although trends in Brazil have shown a consistent decline in beans consumption throughout the last decades [26,27]. Fruit content in the optimized diets increased from 216g to 341g; however, it can be argued that the observed consumption was quite low; thus, the high increase compensates for insufficient baseline consumption. Likewise, it might be challenging to increase the nut consumption from 0.2g/day to 10g/day for the population. Considerable changes in some foods are necessary due to the fact that the current diet that fails to meet most nutrient requirements. Moreover, the changes encompass more than only cost and food preferences, including aspects of convenience [28]. One of the main changes included substantial increases in fruit and vegetable consumption, which implies at least weekly visits to the market due to shorter shelf-life in comparison to other foods. In addition, cooking hard vegetables and beans requires more time and energy (electricity or gas) to prepare. Commonly observed nutrient inadequacies were registered in both low- and high-income households, that is, even disregarding costs, changes to fulfill nutrient adequacy were substantial among different households in the Brazilian population [17].

Second, the cost increased by 20% in nutritionally adequate optimized diets, considering the diet with higher similarity to the current food consumption. It is worth noting that under higher flexibility of diet composition, there were diets with lower costs. However, costs were

associated with changes in diet composition in a non-linear pattern: small divergence of optimized diets in comparison to observed diets resulted in lower differences in costs. Similar non-linear inverse relationship between cost and deviation from the observed diet was found in a study with French women: to meet nutritional constraints, the minimum departure from the observed diet was 495% at a cost of 4.99 euros, but to achieve nutritional constraints at the lowest possible cost (i.e., 3.18 euros), the deviation markedly increased to 2,870% [18].

In the present study, the minimum cost required to reach nutritional adequacy was higher than the cost of the observed diet, but the magnitude of this increase was greater for the lowest income groups. In this sense, irrespective of how much people in lower income groups tolerate substantial changes in the food choices, the adequacy probably will not be attained without increments in cost. Thus, nutritional counseling will not be sufficient unless poorer people are able to increase their food budget. In the lowest income level, the budget for food should increase on average by US$0.10 (BRL0.23) per capita per day, irrespective of the diet acceptability. On the other hand, in the highest income groups, adequacy can be attained at no cost or even with a reduction of the food budget. Compared to the French studies, the cost increment in order to reach nutritional adequacy corresponded to +3.2% of the mean observed diet cost; and similarly to our results, the cost increment was higher in the lower income quintiles [20].

Third, we have shown that calcium recommendation is the costliest and the more demanding in terms of changes in food consumption. Deviation from the observed diet was substantially lower when the constraint for calcium was removed, in comparison to the solutions excluding constraints of other nutrients. Moreover, the impact of removing each of the nutrient constraints, with an exception for calcium, was similar in terms of cost and acceptability in relation to the full nutrient-constrained models. Calcium is described as one of the nutrients with higher inadequacy in many populations worldwide [4,5]; in Brazil, the prevalence of inadequacy found is more than 80% [6,17].

The difficulty to meet calcium requirement was already addressed in a previous study in Brazil using linear programming [23]. Relaxing calcium constraint made the optimized diet cheaper than the observed cost and more realistic. Moreover, the consequence of not meeting dietary calcium requirement is controversy. In a recent review on calcium intake and risk of bone fractures, the conclusions indicated no evidence that increasing calcium intake from dietary sources prevents fractures, and authors argued that recommendations of calcium intake required for bone health have been developed based on findings of early Ca-balance studies, which has never been shown to be associated with fracture risk [29]. In this line, the WHO report on nutrition and chronic disease states that a minimum of 400-500mg of calcium intake would be sufficient to prevent osteoporosis in countries with a high fracture incidence [24], similar to what was obtained in the calcium constraint-free model (505mg).

Once our approach takes into account the actual diet cost and food habits across de country, these results may provide insights into establishing food guides and policies potentially more effective. For example, food guides and polices could explicitly encourage higher consumption of foods identified in the models (e.g. beans, dairy, FV). It also can be useful to identify regional variations in the healthy diets within the country that should be accounted for in the food guides. Moreover, considering the low consumption of calcium and the wide acceptability of dairy foods in the Brazilian population, observed particularly during periods of affluence in the country [30], public policies towards incentives for dairy consumption may benefit lower-income individuals nationwide (e.g., policies to reduce price or subsidies for families with children).

Choices of inputs and constraints imposed in the optimization models need further considerations. The set of recommendations adopted (Dietary Reference Intakes) refers to populations of the USA and Canada, taking into account factors as geography, anthropometry, and

nutrient bioavailability from these populations. In this sense, we opted to exclude constraints for vitamin D and E in the models proposed since vitamin D requirement assumes minimal sun exposition because of high imprecision in sunlight exposure due to skin pigmentation, genetics, latitude, use of sunscreens, and cultural differences in dressing habits, among other factors [31]. It is likely that tropical countries, such as Brazil, need less vitamin D from diet than countries from higher latitude, but to date, we cannot know how much the solar exposition by itself suffices the physiological vitamin D needs [32]. In addition, both DRI and WHO reports stated that there is insufficient information to define indicators for vitamin E adequacy, and they are mainly based on the mean intakes observed in the USA and other European countries [32,33].

The mean observed diet cost was derived from the actual food consumption reported in the food records. A validation study prior to the NDS collection estimated the underreporting energy intake of about 30% comparing with energy expenditure from double-labeled water [34]. Thus, the distances between the observed and the optimized food quantities in the present study are probably overestimated. Furthermore, our results rely on the assumption that smaller differences between observed and optimized diets (in grams of food) ensure improved acceptability of the food changes.

We are also assuming that all food prices are those as purchased in markets or street vendors after adjusting for cooking factors and removing non-edible portions. It may underestimate the diet cost when the food price refers to a food that is assumed to be prepared at home but it is purchased ready to eat, such as meals at the restaurants. On the other hand, it may overestimate the diet cost when the food price refers to a food that is assumed to be purchased ready to eat, but it is prepared at home, such as cakes and sandwiches. It is unknown, however, how much these opposed scenarios are balanced out.

Finally, it is important to point out that the data collection was performed approximately ten years ago. However, it is also the most recent nationwide data on food consumption and prices to date. Additionally, this represents an innovative study that takes into account the populational strata in the diet modeling, accommodating actual food intakes and prices within nationally representative sets of households marked by geographic and socioeconomic homogeneity within the clusters. To our knowledge, it is the unique study assessing food prices and consumption in the same household in the same period of collection. It is especially important in the context of a large and heterogeneous country such as Brazil. For example, the mean daily fish intake varies from 6.8g in Southern to 95g in Northern Brazil.

In conclusion, the study shows that nutritionally adequate diets are possible, usually at higher costs in comparison to the observed dietary patterns in Brazil; and that cost increments presented lower impacts on household budgets in case of substantial changes in the current food patterns, being especially demanding in the lowest income population group.

## Supporting information

**S1 Appendix. Food categorization.**
(XLSX)

## Author Contributions

**Conceptualization:** Eliseu Verly, Jr, Nicole Darmon, Rosely Sichieri, Flavia Mori Sarti.

**Formal analysis:** Eliseu Verly, Jr.

**Funding acquisition:** Eliseu Verly, Jr.

**Methodology:** Eliseu Verly, Jr, Flavia Mori Sarti.

**Supervision:** Nicole Darmon.

**Writing – original draft:** Eliseu Verly, Jr.

**Writing – review & editing:** Nicole Darmon, Rosely Sichieri, Flavia Mori Sarti.

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
