## [Decision Letter · Decision Letter 0]

5 Nov 2019

PONE-D-19-17275

Reaching culturally acceptable and adequate diets at lowest cost increment according to income level in Brazilian households

PLOS ONE

Dear Prof. Verly-Jr,

Thank you for submitting your manuscript to PLOS ONE. After careful consideration, we feel that it has merit but does not fully meet PLOS ONE’s publication criteria as it currently stands. Therefore, we invite you to submit a revised version of the manuscript that addresses the points raised during the review process.

Your manuscript is an important contribution to the literature and both Reviewers were positive overall. There were some editorial suggestions that indicate a need for detailed proof-reading and a number of questions for further clarity of reporting. Please address these concerns in a revised manuscript for consideration for publication.

We would appreciate receiving your revised manuscript by Dec 20 2019 11:59PM. To enhance the reproducibility of your results, we recommend that if applicable you deposit your laboratory protocols in protocols.io, where a protocol can be assigned its own identifier (DOI) such that it can be cited independently in the future. For instructions see: http://journals.plos.org/plosone/s/submission-guidelines#loc-laboratory-protocols

We look forward to receiving your revised manuscript.

Kind regards,

Annalijn I Conklin, M.Sc., M.P.H., Ph.D.

Academic Editor

PLOS ONE

Journal Requirements:

Additional Editor Comments (if provided):

Your paper was well received by the Reviewers, both of whom raised a number of comments to consider to further improve the manuscript. Please address the suggestions, including editorial recommendations for clarity of communication.

Reviewers' comments:

Reviewer's Responses to Questions

**Comments to the Author**

1. Is the manuscript technically sound, and do the data support the conclusions?

Reviewer #1: Yes

Reviewer #2: Yes

2. Has the statistical analysis been performed appropriately and rigorously? 

Reviewer #1: Yes

Reviewer #2: Yes

3. Have the authors made all data underlying the findings in their manuscript fully available?

Reviewer #1: Yes

Reviewer #2: Yes

4. Is the manuscript presented in an intelligible fashion and written in standard English?

Reviewer #1: Yes

Reviewer #2: Yes

5. Review Comments to the Author

Reviewer #1: I enjoyed reviewing this paper utilizing linear programming (LP) in Brazil. The analysis is well-described and the discussion section is relevant without being too long and unwieldy. With that said, I have a handful of comments that would improve the paper and increase the reader understanding of LP method/techniques and limitations.

Scientific comments

Can you clarify the method of dietary data collection? Was it a 24-hour recall or food diary/food record? It is unclear as currently written.

Can slightly more detail be provide regarding the aggregation of data from 305 unique foods to the 26 final food groups? This is a critical step in the process and needs additional explanation.

Does the food price database assume all foods are consumed from stores/supermarkets, as opposed to at restaurants or other value-added products? This has important implications for the differences in cost by socioeconomic strata.

It appears that the food repertoire increased in the optimized models (e.g., an increase in dietary diversity). Could the authors discuss this in slightly more detail, with consideration for the additional cost implications of a more diverse diet (e.g., additional shopping, storage, food wastage, etc).

In the discussion the authors mention that the calcium analysis was shown/discussed but I did not see this information in any figures/tables or described in the Results section. This finding was of much interest and it would be helpful to see slightly more information about this in the results section.

In some cases presenting the diet cost changes as a proportionate increase may increase the saliency? Increasing by $0.78 may not sound like too much but it is a 53% increase, which sounds very different.

Can units be added to Table 3/4 ; assume grams/d but could be more clear.

When discussing deviations for the first time would make clear that these could be +/-. It is a bit unclear in some places.

I was curious that cake went up in the optimized diets. Do the authors have any thoughts as to why this is? Not sure it needs to be noted in the manuscript but probably a question many readers will have.

The second to last sentence of the abstract is difficult to follow; I get what the authors are saying after reviewing the two figures but the language could be more clear.

Since PLOS One does not charge for color images (if I remember correctly), it would be great if Figure 2 could be color. It is a bit challenging to tell the different economic strata from one another.

Lastly, the term “geographic information strata” makes sense but use of the term GIS is quite confusing as people will think you are talking about geographic information systems. Can a different acronym be used?

Editorial comments

While the paper is very clear to follow, there are numerous places where the grammar is not ideal. I have made an effort

to point out a few of these issues, but my comments are not exhaustive. One of the main issues is with use of plural terms

when not appropriate or vice versa.

Abstract, “seafood” not “seafoods”.

Page 4, line 3-4: “there was need to greater changes” could be “there was a need for greater changes”.

Page 8, line 3: “sets” not “set”

Page 11, line 15: Suggest: “we ranked the solutions in ascending order by …”

Page 19, line 5, delete “Besides”

Page 19, line 11: The initial structure of this paragraph is a bit odd with the use of colon and then not a specific list.

Page 20, line 2: add “for” before insufficient. Suggest “might be challenging to increase nut consumption.”

Page 20, line 4: suggest “are necessary due to the fact that the current diet fails…”

Page 20, line 6: Suggest “increases in fruit”

Page 20, line 8: Suggest “requires more time and energy (e.g., electricity or gas) to prepare”

Page 20, line 13: “noting” not “noticing”

Page 20, line 25: suggest “unless poor people are able to increase their food budget”

Page 21, line 3: suggest “Compared to French studies, the cost increment…”

Page 21, line 13: delete “nationwide studies”

Page 22, line 4: suggest “we opted to exclude constraints for…”

Page 22, line 12: suggest “intakes”.

Reviewer #2: • General comment: your manuscript has high use of words such as “the” and “to” which tend to promote the passive voice. Though this is not a fatal flaw, switching to a more active voice truly helps the reader understand your sentences. Consider switching your tone to more active voice where possible. I have done a few suggested changes in the manuscript.

• Similarly, I have taken the liberty of proposing sentence structure changes throughout the manuscript. I understand that English might not be the first language of the authors so I hope it is not offensive that I have addressed some misspellings and sentence structural issues.

• The short title is easier to follow

• Drewnowski is the leading authority on most of those issues. He has a paper about the nutrient health index of foods and their affordability which might be of interest to your paper: Drewnowski, A. (2010). The Nutrient Rich Foods Index helps to identify healthy, affordable foods. The American journal of clinical nutrition, 91(4), 1095S-1101S.

• Page 3 Line 13: This sentence is quite unclear. What do you mean? 21% of households make twice the minimum wage? Is the minimum wage hourly in Brazil, similarly to the USA? Please explain. Also specify the year where these statistics come from and what the source is.

• Page 3 Line 16: This also is not a causal link, which you should make clear here.

• Page 5, line 2: You state the data collection information can be found “elsewhere” which is a bit lackluster. Would you please give a brief explanation of how the data are collected? Nutritional data, as well as food purchase data, are often problematic and their reliability depends on how the data are collected. It would alleviate any data consistency concerns if you could explain how the data are collected.

• Page 5, Line 16: I do not understand what you mean by this. Maybe include a short example so the reader can understand what you are doing and why you are doing it. Are you doing this so that you can observe each stratum over time, converting your data into a pseudo-panel format? This is what it seems like to me but it is unclear.

• Page 6, line 6: again most scientists working with pricing data would be wary of the reliability of self-reported prices, which seems to be what you are using for this study. The process through which the data are collected highly determines how reliable the price data are. If in your case you do not deem this to be an issue, please explain why. If you realize this is an issue but cannot get around it, please explain that in this section.

• Page 6 line 21: you should provide us a short table of the food categories you include and which you exclude so the reader understands what is going into your model.

• Page 8: good discussion on constraints.

• Page 14, line 13: interesting finding.

• Discussion is adequate but it would be helpful to include more policy relevant recommendations. What should public policy do with your findings? How can the findings help provide better nutrition for low income populations since the cost was higher to them to read a nutritionally adequate diet. Find some studies that might support your recommendations.

6. PLOS authors have the option to publish the peer review history of their article (what does this mean?). If published, this will include your full peer review and any attached files.

Reviewer #1: Yes: Colin D. Rehm

Reviewer #2: Yes: Dominique J. Rolando

---

## [Author Response · Author response to Decision Letter 0]

20 Dec 2019

Dear Editor and reviewees, we thank for your time to read and revise the manuscript, we are sure that all the comments and corrections gave us valuable assistance in order to better present our research. 

We made the most of this review and did some modifications in the original submission that were not directly included in the reviewers’ comments. Due to an ongoing parallel study using the same data set, where we re-discussed the analysis protocol such as that used in this manuscript, we decided to:

- revise the food classification, deploying as much as possible the recipes into single foods, and then classify them in their corresponding food groups. Ex.: smoothies were deployed into milk and fruit.

- replace the acceptability constraints derived from the percentiles by the mean portion size, and allowing a progressive increase by every 1gram in case of model infeasibility, until finding a feasible solution. This led to very similar results; maybe the most important difference concerning the fact that in this version the diet cost was cheaper compared to the original submission. We considered it as an important improvement in the analysis. In general, results, discussion, and conclusion did not change.

- work with a reduced number of strata (now referred as to Geographic economic strata - GES). In doing so, we got more precise estimates of the food consumption per unit of analysis, while preserving the socioeconomic, food patterns, and food prices variation across the GESs.

- replace the original Table 4 by a more intuitive and illustrative graph (Figure 2).

We hope you find this version clearer and appreciate our answer to your comments. 

Reviewer #1: I enjoyed reviewing this paper utilizing linear programming (LP) in Brazil. The analysis is well-described and the discussion section is relevant without being too long and unwieldy. With that said, I have a handful of comments that would improve the paper and increase the reader understanding of LP method/techniques and limitations.

Scientific comments

Can you clarify the method of dietary data collection? Was it a 24-hour recall or food diary/food record? It is unclear as currently written.

Answer: The method of dietary data collection was the food records. We included this information in the manuscript.

Can slightly more detail be provide regarding the aggregation of data from 305 unique foods to the 26 final food groups? This is a critical step in the process and needs additional explanation.

Answer: A more detailed sentence was included in the manuscript. In addition, we added an online supplementary file with the foods and food groups used in the categorization.

Does the food price database assume all foods are consumed from stores/supermarkets, as opposed to at restaurants or other value-added products? This has important implications for the differences in cost by socioeconomic strata.

Answer: The Household Budget Survey collects the price of every product purchased, whatever the place, but used/consumed at home. In most cases, the prices refer to markets, but it also records foods purchased from restaurants, snack houses, street vendors, etc, when these foods were strictly consumed at home. A sentence clarifying it was included in the discussion.

It appears that the food repertoire increased in the optimized models (e.g., an increase in dietary diversity). Could the authors discuss this in slightly more detail, with consideration for the additional cost implications of a more diverse diet (e.g., additional shopping, storage, food wastage, etc).

Answer: The increase in dietary diversity consisted of an alternative to reach nutritional goals. The cost increment was partly due to the modifications in the food quantities, and partly because of the inclusion of new foods. In fact, the increase in the food repertoire did not add cost beyond that needed to reach diet adequacy, otherwise, the model would not select these foods when minimizing the diet cost. Not allowing new foods would probably either increase the diet cost or increase the changes in the diet.

In the discussion the authors mention that the calcium analysis was shown/discussed but I did not see this information in any figures/tables or described in the Results section. This finding was of much interest and it would be helpful to see slightly more information about this in the results section.

Answer: These numbers are presented in the topic “Impact of each nutrient adequacy on the cost and deviation” in the Result section. They were not presented in figures or tables.

In some cases presenting the diet cost changes as a proportionate increase may increase the saliency? Increasing by $0.78 may not sound like too much but it is a 53% increase, which sounds very different.

Answer: Thanks, we added a column in the table 2 with the relative increase.

Can units be added to Table 3/4 ; assume grams/d but could be more clear.

Answer: Thanks, we included this information in the table 3. Table 4 was replaced by a graph.

When discussing deviations for the first time would make clear that these could be +/-. It is a bit unclear in some places.

Answer: Thanks, we made it clearer in this version.

I was curious that cake went up in the optimized diets. Do the authors have any thoughts as to why this is? Not sure it needs to be noted in the manuscript but probably a question many readers will have.

Answer: In this reanalysis, the amount of cake was reduced in the optimized diets.

The second to last sentence of the abstract is difficult to follow; I get what the authors are saying after reviewing the two figures but the language could be more clear.

Answer: Thanks. We revised the abstract.

Since PLOS One does not charge for color images (if I remember correctly), it would be great if Figure 2 could be color. It is a bit challenging to tell the different economic strata from one another.

Answer: Thanks for this suggestion. The figure was changed.

Lastly, the term “geographic information strata” makes sense but use of the term GIS is quite confusing as people will think you are talking about geographic information systems. Can a different acronym be used?

Answer: Thanks, we agree with the reviewer. We replaced the term “geographic-income strata” by “geographic-economic strata”, and then the acronym turned to “GES”. 

Editorial comments

While the paper is very clear to follow, there are numerous places where the grammar is not ideal. I have made an effort to point out a few of these issues, but my comments are not exhaustive. One of the main issues is with use of plural terms when not appropriate or vice versa.

Abstract, “seafood” not “seafoods”.

Page 4, line 3-4: “there was need to greater changes” could be “there was a need for greater changes”.

Page 8, line 3: “sets” not “set”

Page 11, line 15: Suggest: “we ranked the solutions in ascending order by …”

Page 19, line 5, delete “Besides”

Page 19, line 11: The initial structure of this paragraph is a bit odd with the use of colon and then not a specific list.

Page 20, line 2: add “for” before insufficient. Suggest “might be challenging to increase nut consumption.”

Page 20, line 4: suggest “are necessary due to the fact that the current diet fails…”

Page 20, line 6: Suggest “increases in fruit”

Page 20, line 8: Suggest “requires more time and energy (e.g., electricity or gas) to prepare”

Page 20, line 13: “noting” not “noticing”

Page 20, line 25: suggest “unless poor people are able to increase their food budget”

Page 21, line 3: suggest “Compared to French studies, the cost increment…”

Page 21, line 13: delete “nationwide studies”

Page 22, line 4: suggest “we opted to exclude constraints for…”

Page 22, line 12: suggest “intakes”.

Answer: Thank you very much for your effort to correct some of the grammar mistakes. In addition to your corrections, we revised the manuscript with the assistance of a native English speaker.

Reviewer #2: • General comment: your manuscript has high use of words such as “the” and “to” which tend to promote the passive voice. Though this is not a fatal flaw, switching to a more active voice truly helps the reader understand your sentences. Consider switching your tone to more active voice where possible. I have done a few suggested changes in the manuscript.

Answer: Thank you very much for your suggestions. With the assistance of a native English speaker, we changed some sentences to the active voice. 

• Similarly, I have taken the liberty of proposing sentence structure changes throughout the manuscript. I understand that English might not be the first language of the authors so I hope it is not offensive that I have addressed some misspellings and sentence structural issues.

Answer: Thanks a lot. We really appreciate your corrections.

• The short title is easier to follow

• Drewnowski is the leading authority on most of those issues. He has a paper about the nutrient health index of foods and their affordability which might be of interest to your paper: Drewnowski, A. (2010). The Nutrient Rich Foods Index helps to identify healthy, affordable foods. The American journal of clinical nutrition, 91(4), 1095S-1101S.

Answer: Thanks for this suggestion, it is indeed a very interesting paper that we had already read previously. We had not considered papers on nutrient profile index because they usually focus on individual foods without taking menus, or the overall diet quality into account. According to the author, “this research needs to be extended to finer food subgroups and applied further using diet optimization techniques to construct affordable healthful diets”. Although it is of overall interest in this field, particularly to our manuscript, we are not sure that discussing the findings of different scopes and approaches would make the discussion more informative.

• Page 3 Line 13: This sentence is quite unclear. What do you mean? 21% of households make twice the minimum wage? Is the minimum wage hourly in Brazil, similarly to the USA? Please explain. Also specify the year where these statistics come from and what the source is.

Answer: We corrected the sentence to clarify its meaning and included the source of information.

• Page 3 Line 16: This also is not a causal link, which you should make clear here.

Answer: Thanks. We changed this sentence in order to make it clearer.

• Page 5, line 2: You state the data collection information can be found “elsewhere” which is a bit lackluster. Would you please give a brief explanation of how the data are collected? Nutritional data, as well as food purchase data, are often problematic and their reliability depends on how the data are collected. It would alleviate any data consistency concerns if you could explain how the data are collected.

Answer: We included more information on the data collection. Yet, it is still summarized in order to not excessively extend the manuscript length. 

• Page 5, Line 16: I do not understand what you mean by this. Maybe include a short example so the reader can understand what you are doing and why you are doing it. Are you doing this so that you can observe each stratum over time, converting your data into a pseudo-panel format? This is what it seems like to me but it is unclear.

Answer: We edited this sentence to make it clearer.

• Page 6, line 6: again most scientists working with pricing data would be wary of the reliability of self-reported prices, which seems to be what you are using for this study. The process through which the data are collected highly determines how reliable the price data are. If in your case you do not deem this to be an issue, please explain why. If you realize this is an issue but cannot get around it, please explain that in this section.

Answer: Thanks, we added this sentence in the methods section: “Data referring to prices are indirectly inferred: individuals report expenditures and amounts, and prices are calculated using the division of expenditure per item in relation to its respective amount. The information on expenditures is collected using both self-reported information and receipts presented by the individuals interviewed, which are checked by the interviewer in order to ensure its reliability”.

• Page 6 line 21: you should provide us a short table of the food categories you include and which you exclude so the reader understands what is going into your model.

Answer: All the food items used in the models were reported in the dietary survey. The items excluded are stated in the manuscript: coffee and tea, and alcoholic beverages. The items included in the analysis are presented in Table 2 of the Results section. We included an online supplementary file with the foods and food groups used in the categorization.

• Page 8: good discussion on constraints.

Answer: Thanks.

• Page 14, line 13: interesting finding.

Answer: Thanks.

• Discussion is adequate but it would be helpful to include more policy relevant recommendations. What should public policy do with your findings? How can the findings help provide better nutrition for low income populations since the cost was higher to them to read a nutritionally adequate diet. Find some studies that might support your recommendations.

Answer: Thanks for this comment. This is somewhat discussed, especially in the first paragraph where we stated “these results provide insights in establishing food guides and policies potentially more effective using optimization tools, especially in developing countries. For example, the results point to a possibility of adopting incentives in production and/or distribution of certain key foods to reduce prices promoting higher intake that would potentially result in higher diet quality with a greater chance of adherence by the population.” We argue that more detailed or specific actions deriving from these findings should involve agronomy, economy, and other sciences, and it is out of the scope of this manuscript.

---

## [Editor Report · Decision Letter 1]

24 Jan 2020

PONE-D-19-17275R1

Reaching culturally acceptable and adequate diets at the lowest cost increment according to income level in Brazilian households

PLOS ONE

Dear Prof. Verly-Jr,

Thank you for submitting your manuscript to PLOS ONE. After careful consideration, we feel that it has merit but does not fully meet PLOS ONE’s publication criteria as it currently stands. Therefore, we invite you to submit a revised version of the manuscript that addresses the points raised during the review process.

Thank you for responding to the Reviewers’ comments. For the most part, they are adequately addressed. However, I agree with the Reviewer that it would be helpful to include more policy relevant recommendations as the current text is vague and vacuous. The findings seem to indicate that the “certain key foods” that have the highest increase in price from observed to optimized diet may be the best target for incentives, (i.e. eggs and dairy products). Could the authors offer more concrete recommendations for how results from optimization tools offer insights in establishing food guides and policies?

There are also a few errors that were introduced with the new text: 

p.6, line5 

This sentence does not make sense: “It was reported 305 different types of food items, which 6 comprised similar items with distinct subtypes or preparation methods, for instance, different types 7 of banana, or different preparation of red meat (boiled, roasted, grilled, etc.).” 

Page 9. Line 16. M

issing word (“the model”) between “till” and “find a feasible solution”

We would appreciate receiving your revised manuscript by Mar 09 2020 11:59PM. To enhance the reproducibility of your results, we recommend that if applicable you deposit your laboratory protocols in protocols.io, where a protocol can be assigned its own identifier (DOI) such that it can be cited independently in the future. For instructions see: http://journals.plos.org/plosone/s/submission-guidelines#loc-laboratory-protocols

We look forward to receiving your revised manuscript.

Kind regards,

Annalijn I Conklin, M.Sc., M.P.H., Ph.D.

Academic Editor

PLOS ONE

Additional Editor Comments (if provided):

Thank you for responding to the Reviewers’ comments. For the most part, they are adequately addressed. However, I agree with the Reviewer that it would be helpful to include more policy relevant recommendations as the current text is vague and vacuous. The findings seem to indicate that the “certain key foods” that have the highest increase in price from observed to optimized diet may be the best target for incentives, (i.e. eggs and dairy products). Could the authors offer more concrete recommendations for how results from optimization tools offer insights in establishing food guides and policies?

There are also a few errors that were introduced with the new text: 

p.6, line5 

This sentence does not make sense: “It was reported 305 different types of food items, which 6 comprised similar items with distinct subtypes or preparation methods, for instance, different types 7 of banana, or different preparation of red meat (boiled, roasted, grilled, etc.).” 

Page 9. Line 16. 

Missing word (“the model”) between “till” and “find a feasible solution”

---

## [Author Response · Author response to Decision Letter 1]

4 Feb 2020

Dear Editor and reviewers, we thank for your time to read and revise the manuscript.

Editor comments:

Thank you for responding to the Reviewers’ comments. For the most part, they are adequately addressed. However, I agree with the Reviewer that it would be helpful to include more policy relevant recommendations as the current text is vague and vacuous. The findings seem to indicate that the “certain key foods” that have the highest increase in price from observed to optimized diet may be the best target for incentives, (i.e. eggs and dairy products). Could the authors offer more concrete recommendations for how results from optimization tools offer insights in establishing food guides and policies?

Answer: we accepted your suggestion and included a sentence in the manuscript.

There are also a few errors that were introduced with the new text:

p.6, line5

This sentence does not make sense: “It was reported 305 different types of food items, which 6 comprised similar items with distinct subtypes or preparation methods, for instance, different types 7 of banana, or different preparation of red meat (boiled, roasted, grilled, etc.).”

Answer: Thanks for noting this mistake. We replaced the sentence by: “It was reported 305 different food items, most of them were aggregated into a single food, for example, different types of banana into banana, or different preparation of red meat (boiled, roasted, grilled, etc.) into red meat. The aggregation resulted in a list of 102 foods.”

Page 9. Line 16.

Missing word (“the model”) between “till” and “find a feasible solution”

Answer: Thanks, it was corrected in the manuscript.

---

## [Editor Report · Decision Letter 2]

7 Feb 2020

Reaching culturally acceptable and adequate diets at the lowest cost increment according to income level in Brazilian households

PONE-D-19-17275R2

Dear Dr. Verly-Jr,

We are pleased to inform you that your manuscript has been judged scientifically suitable for publication and will be formally accepted for publication once it complies with all outstanding technical requirements.

With kind regards,

Annalijn I Conklin, M.Sc., M.P.H., Ph.D.

Academic Editor

PLOS ONE

Additional Editor Comments (optional):

Thank you for your additional edits to this manuscript.
---

## [Editor Report · Acceptance letter]

13 Feb 2020

PONE-D-19-17275R2 

Reaching culturally acceptable and adequate diets at the lowest cost increment according to income level in Brazilian households 

Dear Dr. Verly-Jr:

I am pleased to inform you that your manuscript has been deemed suitable for publication in PLOS ONE. Congratulations! Your manuscript is now with our production department. 

With kind regards,

on behalf of

Dr. Annalijn I Conklin 

Academic Editor

PLOS ONE